# ROBUSTNESS OF TRUSS DECOMPOSITION AND IMPLICATIONS FOR GNN-BASED EDGE CLASSIFICATION

## ABSTRACT

Truss decomposition is an effective and practical algorithm for dense subgraph discovery. However, it is sensitive to the changes in the graph: dropping a few edges or a bit of noise can drastically impact the truss numbers of the edges. It is of practical importance to understand and characterize the robustness of truss decomposition. In this work, we study and utilize the robustness of truss decomposition in an edge-driven way. We propose to construct a dependency graph among edges to denote the impact of an edge's removal on the neighboring edges. By using the dependency graph, we introduce three measures to capture the diverse and unique properties of the edges. We provide theoretical findings and design an efficient algorithm to compute the dependency graph faster than the naive baseline. We also show that our new edge-based truss robustness measures capture intrinsic graph structures and have the potential to unearth peculiar differences that can help with various downstream tasks, such as edge classification. We integrate our measures into the state-of-the-art GNN for edge classification and demonstrate improved performance on multi-class datasets. The overhead of computing our edge-based measures is insignificant when compared to the training time. We believe that utilizing edge-based truss and robustness measures can further be helpful in edge-driven downstream tasks.

## 1    INTRODUCTION

Dense subgraphs are found to be useful in various applications such as anomaly detection, visualization, and clustering (Shin et al., 2018; Liu & Sarıyüce, 2020; Alvarez-Hamelin et al., 2005; Gibson et al., 2005). Among many methods, truss decomposition has attracted particular interest due to its effectiveness and practicality (Sariyuce et al., 2015; Sariyüce et al., 2017; Wang & Cheng, 2012; Fang et al., 2020; Huang et al., 2016; Wang & Cheng, 2012). $k$-truss is the maximal subgraph in which every edge is contained in at least $k$ triangles (Cohen, 2008). Truss number (or trussness) of an edge is the largest $k$ for which the edge is part of a $k$-truss. $k$-truss is inspired by the concept of $k$-core—the maximal subgraph in which every node has a degree of at least $k$ (Seidman, 1983)—and it is a superior alternative to $k$-core, outperforming it in various applications such as internet AS-level analysis (Gregori et al., 2013; Carmi et al., 2007; Alvarez-Hamelin et al., 2005), visualization (Healy et al., 2008; Colomer-de Simón et al., 2013), and community detection (Sariyuce et al., 2015; Sariyüce et al., 2017).

Despite its broad applicability, $k$-trusses are notoriously sensitive to changes in the graph (Chen et al., 2021; 2022; Zhu et al., 2019). Even removing a few edges or adding some noise can have cascading effects and influence the truss numbers of surrounding edges. Chen et al. (2021) demonstrated that even minor edge removals can significantly impact truss-based communities, emphasizing the need for taking truss robustness into account in downstream tasks. While core decomposition robustness has been extensively studied (Zhou et al., 2021; Hossain et al., 2023), truss robustness remains relatively unexplored, despite its superiority in various tasks across diverse data types (Gregori et al., 2013). Considering the wide application space of $k$-trusses, it is of utmost importance to comprehend and characterize the robustness of truss decomposition on the edge-level. Identifying which edges are susceptible to nearby changes or determining the edges whose removal has a more pronounced impact on neighboring edges becomes crucial in this context. To the best of our knowledge, there

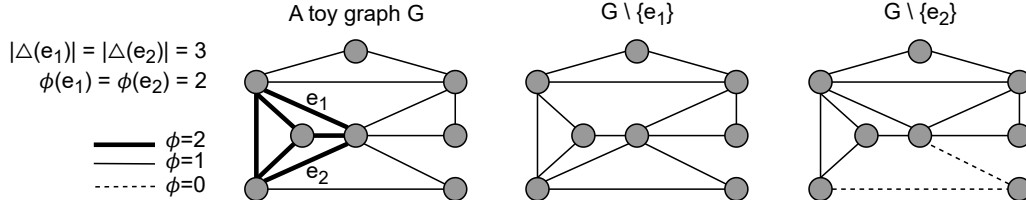

Figure 1: A toy graph showing the nuances captured by the robustness of truss numbers. $\phi$ denotes truss number and $|\triangle(e)|$ is the number of triangles that contain $e$. $e_1$ and $e_2$ have the same triangle counts and same truss numbers in the toy graph on the left, however their role in the graph structure are not the same: removing each edge has unique impact on the surrounding edges—truss numbers are different after removing each, as denoted on the middle and right.

is no study that quantifies edge-based truss robustness. The increasing importance of edge-driven methods in graph analysis (Benson et al., 2016; Bick et al., 2023) underscores the need for robust edge-based measures that can capture intricate graph structures. In scenarios where edge properties are key, such as anomaly detection and edge classification, truss numbers as well as their robustness qualities could offer unique insights. For instance, Li et al. (2024) recently proposed a $k$-truss based temporal graph convolutional network to capture both intricate topology and temporal dependencies in graphs, demonstrating the potential of truss-based approaches in enhancing GNN performance.

To exemplify the nuances truss robustness captures, we provide a toy graph in Figure 1 Edges $e_1$ and $e_2$ have the same triangle counts and same truss numbers in the toy graph (left). However, removing them has different impacts on the truss number of other edges. Removing $e_1$ solely impacts the edges with trussness of two, reducing their value to one. Removing $e_2$, however, triggers a broader effect: in addition to decreasing the trussness of all edges with truss number two, it also impacts the two edges with truss number of one. Overall, robustness of the truss numbers helps to distinguish the structural differences around edges, especially when traditional measures such as triangle count and trussness fail, and hence can be useful in downstream tasks like edge classification.

In this study, we characterize, quantify, and utilize the robustness of truss decomposition in an edge-driven way. We propose to model the robustness of truss numbers by establishing a dependency graph among edges—edges in the input graph become nodes in the dependency graph. For any two edges $e$ and $f$ that share a triangle, edge $e$ is defined to be dependent on edge $f$ if the truss number of $e$ decreases after removing $f$, denoted by $f \rightarrow e$ relation. We only focus on the impact of a single-edge removal for simplicity—we hypothesize that multiple-edge removals can be approximated by considering multiple single-edge removals. We remove each edge independently and create its dependency relations to the neighboring edges. By utilizing the dependency graph, we propose three edge-based measures: *Edge Robustness*, *Edge Strength*, and *EdgeRank*, which capture the (inverse of) in-degree, out-degree, and PageRank values of the nodes in the dependency graph, respectively. *Edge Robustness* measures how likely an edge's truss number drops when its neighbors are removed and *Edge Strength* quantifies the edge's potency to alter the truss number of surrounding edges. As computing the dependency graph is computationally costly, we introduce several theoretical findings that enable faster computation. Leveraging these findings, we formulate an efficient algorithm that is 3.74× faster than the naive baseline, on average.

As shown in Figure 1, edge-based truss robustness measures can capture intrinsic graph structures and thus have the potential to reveal nuances between edges in downstream tasks. We provide an early empirical analysis to show the extent of diversity captured by our edge-based measures. To showcase the merits of our novel edge-based measures, we primarily focus on the multi-class edge classification problem (Küçüktunç et al., 2013; Gupta et al., 2014; Behera & Panigrahi, 2015; Duan et al., 2020; Leskovec et al., 2010; Wang et al., 2010) where the class distribution is imbalanced, hence capturing rare classes is challenging. Building on a cutting-edge Graph Neural Network (GNN) designed for edge classification (Wang et al., 2023), we seamlessly integrate our measures to improve the edge classification performance. This integration facilitates the learning of enhanced edge embeddings, leading to up to 3.08% improvement in F1 scores. The improvement is particularly noteworthy in identifying rare classes. Furthermore, we underscore the efficiency of our approach by demonstrating that the computational overhead associated with our edge-based measures is negligible

when compared to the training runtime. We posit that incorporating edge-based truss robustness measures holds the potential for helping edge-driven downstream tasks. Our contributions are summarized as follows:

- We propose the first study to characterize edge-based robustness of the truss decomposition.
- We model the robustness of truss numbers by building a dependency graph among edges and define three edge-based measures based on the dependency graph.
- We introduce theoretical findings and an algorithm to enable practical computation of the dependency graph.
- We perform extensive experiments on real-world networks and demonstrate the effectiveness and efficiency of our measures against other baselines on edge classification task.

## 2 BACKGROUND AND RELATED WORK

In this work, we focus on undirected, unweighted graphs, denoted as $G = (V, E)$ where $V$ represents nodes and $E$ represents edges. Assume $S$ is a subgraph of $G$, $S \subseteq G$. We consider $deg(u, S)$ to denote the degree of $u$ in $S$. If the graph is directed, the notation extends to $deg^+(u, S)$ and $deg^-(u, S)$, representing the out-degree and in-degree of $u$ within $S$, respectively. A triangle in $G$ is a cycle of three nodes and three edges, denoted by $\triangle$. If an edge e is contained in a triangle $\triangle$, we denote it by $e \in \triangle$. $\triangle(e)$ denotes the set of triangles that contain edge $e$ (notation is given in Table 3 at Appendix).

A $k$-truss $G_k$ of an undirected graph $G$ is the maximal triangle-connected subgraph that contains edges that are part of at least $k$ triangles in the subgraph. For each edge $e \in G$, its truss number (or trussness), $\phi(e, G)$, is defined as the maximum $k$ such that $e$ resides in a $k$-truss. We define that two edges are **incident** (or neighbor) if they reside in the same triangle (which is different than the typical definition in graph theory). The set of incident edges of an edge $e$ is denoted as $E(e, G)$. We split the incident edges of $e$ into different sets based on the relative truss numbers, e.g., $\Gamma_<(e, G) = \{e' : e' \in E(e, G) \wedge \phi(e', G) < \phi(e, G)\}$ denotes the neighbors of edge $e$ with smaller truss numbers. The **support** of an edge $e$, denoted by $sup(e, G)$, is the number of triangles containing $e$ in $G$, i.e., $sup(e, G) = |\triangle(e)|$. The **trussness support**, $ts(e, G)$, is the support of edge $e$ in $\phi(e)$-truss (Zhang & Yu, 2019). For each $\triangle \in \phi(e)$-truss, where $e \in \triangle$, there are two incident edges of $e$ contained within $\triangle$. Hence, $ts(e, G) = |\Gamma_\geq(e, \phi(e)\text{-truss})|/2$. Two triangles are **adjacent** if they share a common edge. Two edges are **triangle-connected** if they are incident to each other or there is a series of edges between them that are consecutively incident to each other. A subgraph is **triangle-connected** if every pair of edges in it are triangle-connected. In all notations, we omit $G$ when it is obvious.

**Core and truss robustness.** Existing research on network robustness for dense subgraph discovery leans heavily on $k$-cores (Zhou et al., 2021; Dey et al., 2020; Laishram et al., 2018; Adiga & Vullikanti, 2013; Hossain et al., 2023), while $k$-truss robustness is much less explored (Chen et al., 2021; 2022; Zhu et al., 2019). Traditional methods (Dey et al., 2020; Zhou et al., 2021; Adiga & Vullikanti, 2013; Laishram et al., 2018) have focused on core number changes across the entire graph or within specific subgraphs, neglecting individual node robustness. Hossain et al. (2023) addressed this by proposing Removal Strength (RS), which measure a node's ability to influence or maintain its coreness under edge disruptions. RS has two variants: $RS_{OD}$ measures a node's ability to affect neighboring core numbers upon removal, while $RS_{ID}$ measures its ability to stay within the $k$-core. The authors showed that core robustness measures help to identify critical edges and influential nodes. Despite its usefulness and superiority in various tasks on diverse data types and setups (Gregori et al., 2013; Colomer-de Simón et al., 2013; Orsini et al., 2013; Wang & Cheng, 2012; Zhao & Tung, 2012; Huang et al., 2016), truss robustness has received less attention. Zhu et al. (2019) introduced the concept of using $k$-trusses to evaluate social network stability under edge removals. Building on this, Chen et al. (2021) extended the analysis to consider both edge removals and insertions, addressing community-breaking problems and network robustness. However, no measures have been developed yet to assess the robustness of individual edges within $k$-trusses. In that sense, we propose the first edge-based robustness measures for truss decomposition.

**Edge classification.** Edge-driven methods, which reason about edges instead of nodes, and higher-order techniques are getting more popular due to their novel relation-first approach (Benson et al., 2016; Bick et al., 2023). Edge classification, crucial for tasks like anomaly detection and recommendations (Bielak et al., 2022; Wang et al., 2020; Yang & Xu, 2022), presents unique chal-

lenges. Unlike nodes with inherent features, edges rely on aggregated information from connected nodes. However, existing methods like neighborhood aggregation (Hamilton et al., 2017) or embedding techniques (Grover & Leskovec, 2016; Tang et al., 2015; Menon & Elkan, 2011) struggle to capture complex relationships or handle large-scale graphs effectively. To address this, auto-encoders like Edge2vec (Wang et al., 2020) and AttrE2vec (Bielak et al., 2022) attempt to learn meaningful edge representations by leveraging deep learning architectures or random walks. However, they suffer from high computational costs. GNN-based approaches like SEAL (Zhang & Chen, 2018) and EGNN (Li et al., 2022) focus on the local subgraph structure around an edge for representation learning, achieving good accuracy. More recently, Wang et al. (2023) proposed two novel methods, Topological Edge Representation (TER) and Attributed Edge Representation (AER), specifically designed for efficient and effective edge-wise graph representation learning. TER captures higher-order proximities of edges in a low-dimensional space, significantly improving performance compared to existing approaches. AER augments edge attributes through a carefully-designed feature aggregation scheme, enhancing representation quality for attributed graphs. Combining TER and AER outperforms previous approaches in edge classification. Most of the earlier works on edge classification heavily focus on edge information aggregated from the node attributes or edge features learned from GNNs. Although features like triangle count, trussness (Wang & Cheng, 2012), and node-pair sum of core robustness (Hossain et al., 2023) have been defined earlier, they have not been used for edge classification. Motivated by the intrinsic graph structures captured by our edge-based truss robustness measures, we integrate them into TER and AER to enhance edge classification performance.

## 3 EDGE-BASED TRUSS ROBUSTNESS

We employ a comprehensive approach to capture the robustness of an edge's truss number against edge removals. We determine whether an edge's truss number is dependent or not on any incident edge when the edge is removed from the graph. We consider all the edges and figure out the dependency relationships among them to quantify the robustness of each edge. We focus on the impact of a single-edge removal for simplicity, and create the dependency graph as follows:

**Definition 1** *We define that an edge $e$ is **dependent** on an incident edge $f$, denoted as a relationship $f \rightarrow e$, if $\phi(e)$ decrements after removing the edge $f$. For a given graph $G = (V, E)$, we define the **dependency graph** as a directed graph $G^d = (V^d, E^d)$ where each edge in $G$ is represented as a node in $G^d$ ($V^d = E$) and for each pair of incident edges $e, f$ in $E$, there is a directed edge $(f, e) \in E^d$ if $\phi(e, G \setminus f) < \phi(e, G)$.*

We give an example in Figure 2a. In the toy graph on the top, each edge has a truss number of one. The corresponding dependency graph is given on the bottom. Each edge in the toy graph corresponds to a node in the dependency graph and the edges in the dependency graph represent the dependencies among incident edges in the toy graph. On the bottom, $e_4$ has two in-neighbors ($e_3$ and $e_5$), meaning it is dependent on $e_3$ and $e_5$. For a pair of incident edges in the toy graph, if neither edge is dependent

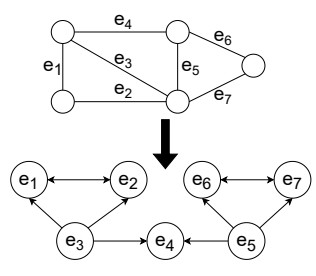

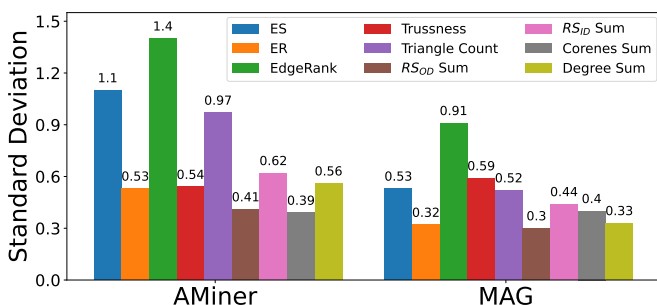

(a) A toy graph (top) and its dependency graph (bottom).

(b) Standard deviation of class-wise mean values for each edge feature in two datasets. A higher standard deviation indicates a more distinguishing edge feature for classification. The high standard deviations of the ES, and EdgeRank features highlight their potential.

Figure 2: Dependency graph example (left) and implications on edge classification (right).

on the other, then no edge appears in the dependency graph—as is the case for $(e_3, e_5)$ which are incident on the left but not connected on the right. Also, if two edges in the toy graph are dependent on each other, there are two edges in both directions in the dependency graph, as for $(e_1, e_2)$.

In-/out-degree of a node in the dependency graph (edge in the original graph, $G$) provides important insights about its truss robustness. A node with a large in-degree is dependent on many of its neighbors, hence removing a nearby edge (in $G$) could reduce its truss number, implying a lower truss robustness. To capture this inversely, we define **Edge Robustness** of a node in $G^d$ to quantify the robustness of the corresponding edge $e \in G$ to retain its truss number upon an incident edge removal:

$$ER(e) = 1/deg^-(e, G^d) \tag{1}$$

The higher a node's out-degree in the dependency graph, the more strength it has to change the other edges' truss numbers. We define **Edge Strength** of a node in $G^d$ to quantify the strength of the corresponding edge $e \in G$ to change the trussness of other edges:

$$ES(e) = deg^+(e, G^d) \tag{2}$$

Our two measures capture the impact of an edge on its direct neighbors' trussness. To capture the higher-order influences of an individual edge, we introduce another measure called **EdgeRank**. EdgeRank of a node in the dependency graph is the PageRank score of it in a modified dependency graph where edge directions are reversed. In the reversed dependency graph ($G^{rd}$), nodes with higher strength (Edge Strength in the dependency graph) have a greater number of incoming links. EdgeRank scores help to discover the nodes that have incoming edges from other nodes with high EdgeRank scores, effectively highlighting their potential to influence the overall truss structure of the network. Below is the formula for **EdgeRank**, where $d$ is the damping factor (default $d = 0.85$), $e'$ represents the incoming neighbors of $e$, and $deg^+(e', G^{rd})$ is the out-degree of $e'$ in the $G^{rd}$:

$$\text{EdgeRank}(e) = (1 - d) + d \cdot \sum_{e' \in \text{InNeighbors}(e)} \text{EdgeRank}(e')/deg^+(e', G^{rd}) \tag{3}$$

**Implications of truss robustness measures for edge classification.** As shown in Figure 1, truss robustness captures intrinsic graph structures that cannot be captured by triangle counts or trussness values. Here we perform an early empirical analysis of our truss robustness measures as well as several graph structural features to see their potential impact on edge classification task. We analyze two multi-class datasets from Wang et al. (2023) (details are given in Table 1). Analysis of other four datasets is given in Figure 5 at Appendix. To find the importance of each edge-based feature, we conduct a thorough analysis of their value distributions across different classes. Generally, a valuable feature exhibits distinct values across classes. The greater this difference, the more effective the feature is for classification. To quantify this difference for each edge feature, we first calculate the average value of the feature across all edges belonging to the same class. This results in a set of average values, one for each class. Then we compute the standard deviation of these class-wise average values. A higher standard deviation indicates greater variation in the feature across classes, suggesting its potential value for edge classification. In essence, a high standard deviation implies that the edge feature can effectively separate different classes. We consider nine features in total, including our three measures, *ES, ER, EdgeRank*, the triangle count and trussness values of edges, and the sum of coreness, degree, $RS_{OD}$, and $RS_{ID}$ values of the endpoints of each edge. All are scaled to the same range. Figure 2b shows the standard deviations for all measures. For each dataset, each bar represents the standard deviation of their values across different classes. We observe high standard deviations for $ES$ and $EdgeRank$ for the AMiner and MAG graphs, suggesting their potential in better distinguishing edge classes. Note that, aside from triangle count and $RS_{ID}$ sum, other variant measures have consistently lower standard deviations for all the datasets. Overall, our edge-based truss robustness measures, particularly $ES$ and $EdgeRank$, have a high potential to be help distinguish classes in the edge classification task.

## 4 AN EFFICIENT ALGORITHM TO COMPUTE THE DEPENDENCY GRAPH

A naive way to compute the dependency graph is to run the incremental truss decomposition algorithm used by Zhang & Yu (2019) for every single edge removal, which will be costly. Here we propose efficient heuristics by using key observations about the truss structure. We begin by introducing a few useful definitions and lemmas about truss number changes upon edge removal.

**Definition 2** *An edge $e \in G$ is defined to be* **vulnerable** *if $\phi(e, G) = |\Gamma_{\geq}(e, G_{\phi(e)})|/2$. For a vulnerable edge $e$, $\Gamma_{\geq}(e, G_{\phi(e)})$ is called the* **sensitive incident edges of** $e$.

Depending on the vulnerable edges and its connectedness within a $k$-truss, we introduce the following definition.

**Definition 3** $k$-**exposed** *is a maximal triangle-connected subgraph of vulnerable edges with the same truss number, $k$. Formally, $S \subseteq G$ is a $k$-exposed if $\forall e \in S$, $\phi(e, G) = k$ and $e$ is a vulnerable edge.*

**Lemma 1** *If a sensitive incident edge $e'$ of a vulnerable edge $e$ is removed, then $\phi(e, G)$ will decrease.*

**Proof 1** *Let us consider that edge $e$ in a $k$-truss subgraph where $\phi(e, G) = |\Gamma_{\geq}(e, G_{\phi(e)})|/2 = k$. This states that edge $e$ has exactly $k * 2$ incident edges (or $ts(e, G) = k$) whose truss numbers are at least $k$. Now, if we delete any sensitive edge $e'$ of $e$ (i.e., $e' \in \Gamma_{\geq}(e, G_{\phi(e)})$), then edge $e$ will have less than $k * 2$ incident edges whose truss numbers are at least $k$ and thus the $ts(e, G)$ will also decrease. As a result, from the definition of truss number, $\phi(e, G \setminus \{e'\})$ will be less than $k$, and lemma holds.*

Focusing on the vulnerable edges and the sensitive incident edges around them can help figuring out the dependency relations in a quicker way.

**Definition 4** *Given a graph $G = (V, E)$ and an edge $e \in E$, the* **subtruss** *of $e$, also denoted as $ST_e$, is a set of edges $e' \in E$ that have $\phi(e') = \phi(e)$ and are reachable from $e$ via a series of adjacent triangles where the common edge $e''$ in every adjacent triangle has truss values $\phi(e'')$ equal to $\phi(e)$.*

**Lemma 2** **(From Huang et al. (2014))** *After removing a single edge, if the truss number of an edge $e$ decreases, then it* **may** *only affect the truss number of other edges in the subtruss of $e$ ($ST_e$).*

**Definition 5** *After removing a single edge, if the truss number of an edge $e$ decreases, the subset of edges $e' \in ST_e$ whose truss number change is called the Truss Changed Edges of $e$ ($TCE_e$).*

Leveraging the previously defined concepts of $k$-exposed and Truss Changed Edges ($TCE$), we introduce the following lemmas to design an efficient algorithm to compute dependency relations.

**Lemma 3** *In a $k$-exposed $S$, if the truss number of an edge $e \in S$ decreases, then the truss number of all the other edges $e' \in S \setminus \{e\}$ will also decrease.*

**Lemma 4** *Given a $k$-exposed $S$ and any two edges $e_1, e_2 \in S$, $TCE_{e_1} = TCE_{e_2}$. TCE of any edge in a $k$-exposed $S$ is denoted by $TCE_S$ (Proofs of Lemma 3 and Lemma 4 are given in Section 7.3 in Appendix).*

**Observation 1** *From Lemma 1 and Definition 5, the truss number of an edge $e'$ will change after the removal of edge $e$ if it satisfies one of the two conditions:*
**1.** *$e'$ is vulnerable and $e$ is a sensitive incident edge of $e'$.*
**2.** *$e'$ is not vulnerable and $e' \in TCE_{e''}$ s.t. $e''$ is vulnerable and $e$ is a sensitive incident edge of $e''$.*

## 4.1 EDGE ROBUSTNESS COMPUTATION (ERC)

Building on our theoretical findings, which culminated in Observation 1, here we provide the **ERC** algorithm (Algorithm 1) for computing edge-based truss robustness measures, $ER$, $ES$, and $EdgeRank$ values of edges, in a given graph.

**ERC algorithm.** To compute the truss robustness measures, we need to construct a *dependency graph* that considers the edges whose removal may affect the truss numbers of their incident edges. While building this *dependency graph* typically involves removing and evaluating each edge, Lemma 4 provides a powerful optimization. Edges within the same $k$-exposed subgraph share the same impact on trussness changes. This allows us to avoid redundant calculations by analyzing the impact of only a single representative edge from a $k$-exposed subgraph and using it for all the other edges.

---

**Algorithm 1: ERC: Edge Robustness Computation** $(G(V, E))$

1 **Input:** $G$ $(V, E)$: graph. **Output:** $ER, ES, EdgeRank$

2 $G^d$ $(V', E') \leftarrow$ empty graph          `// dependency graph`

3 $TCE \leftarrow []$                 `// store TCE_S for any k-exposed, S`

4 $ID_{exp} \leftarrow []$             `// store k-exposed id for each edge`

5 Compute all $k$-trusses of $G$; and $\phi(e)$ and $ts(e)$, where $e \in G$

6 Find all $k$-exposed of $G$, put in $\mathcal{S}_G$ and update $ID_{exp}$

7 **foreach** $k$-exposed $S \in \mathcal{S}_G$ **do** $TCE[S] = \text{TCECompute}(G(V, E), S, \phi, ts)$

8 **foreach** $e \in E$ **do**

9      $TD_e \leftarrow \emptyset$        `// edges whose truss number will decrease after` `removing` $e$

10      **foreach** $e' \in |\Gamma_{\leq}(e, G)|$ **do**

11          **if** $\phi(e', G) = |\Gamma_{\geq}(e', G_{\phi(e')})|/2$ **then**

12              $E'.\text{push}((e, e'))$; **foreach** $e'' \in TCE[ID_{exp}[e']]$ **do** $TD_e.\text{push}(e'')$      `// By` `Lem 1`

13      **foreach** $e' \in |\Gamma_{\leq}(e, G)|$ *s.t.* $e' \in TD_e$ & $(e, e') \notin E'$ **do** $E'.\text{push}((e, e'))$      `// By` `Obs 1`

14 **foreach** $p$ *in* $V'$ **do**

15      $ER(p) \leftarrow \frac{1}{deg^-(p, G^d)}$; $ES(p) \leftarrow deg^+(p, G^d)$; $EdgeRank(p) \leftarrow PageRank(p)$ in reversed $G^d$

16 Return $ER, ES, EdgeRank$

---

In Algorithm 1, we use $TCE$ to store the Truss Changed Edges for all $k$-exposed where the $ID_{exp}$ save the $k$-exposed information of each edge. Leveraging the order-based algorithm of Zhang & Yu (2019), we begin by identifying the $k$-trusses and computing trussness and trussness support of each edge (Line 5). Then by traversing over the edges, we compute the set of all $k$-exposed ($\mathcal{S}_G$) of graph $G$ (Line 6). For each $k$-exposed $S \in \mathcal{S}_G$, we find the $TCE_S$ (stored as $TCE[S]$) by decreasing the truss number of a random edge $e_{rnd} \in S$ (Line 7). Here, to find the $TCE$ of any $k$-exposed $S$, we adapt the algorithm from Zhang & Yu (2019) (explained below). Next, we consider each edge $e \in G$ to compute the dependency information for their incident edges and use $TD_e$ to track all the edges whose $\phi$ will change in $G \setminus \{e\}$. Our algorithms only consider the incident edges $e' \in |\Gamma_{\leq}(e, G)|$ for potential impact when removing any edge $e$. If edge $e$ is a sensitive incident edge of any vulnerable edge $e'$, then removing $e$ will affect $e'$ and we put a directed edge in $G^d$ from $e$ to $e'$ (Line 12). A decrease in $e'$ will also affect the other edges $e'' \in TCE[ID_{exp}[e']]$ (Lemma 4), and this information is accumulated and saved in $TD_e$ for each vulnerable edge $e' \in |\Gamma_{\leq}(e, G)|$ (Line 12). Now, the incident edges ($e'$) of $e$ that are not vulnerable but are affected by some other vulnerable edges will be dependent on $e$ (Observation 1). We check this information (stored in $TD_e$) and put a directed edge in the $G^d$ (Line 13). Finally, we compute and return all three truss robustness measures of the nodes in the dependency graph (edges in the original graph) (Lines 14 to 16).

**Truss changed edges computation (TCECompute).** To compute the Truss Changed Edges of $S$ ($TCE_S$) for any $k$-exposed $S$, we decrease the truss number of only one edge from any $S$ (thanks to Lemma 4). The algorithm is adapted from the order-based algorithm by Zhang & Yu (2019). Different from Zhang & Yu (2019), our approach focuses on strategically decreasing the trussness of an edge rather than removing the edge entirely. Here, we use a queue to manage the set of $TCE_S$ depending on the trussness support. Initially, we choose a random edge $e_{rnd} \in S$ to decrease its trussness, insert it into the queue, update the trussness support ($ts$), and $TCE_S$. Then, we iterate over each edge ($e_{dec}$) from the queue to update the $ts$ of its incident edges and the $TCE_S$. We update the trussness support of incident edges of $e_{dec}$ in the process. Then, for each incident edges $e \in \Gamma_{\leq}(e_{dec}, G)$, we update the queue and $TCE_S$ according to Zhang & Yu (2019). The pseudocode is given in Section 7.4 in Appendix.

**Time complexity.** The time complexity of truss decomposition as well as Line 5 of Algorithm 1 is $O(|E|^{1.5})$ (Wang & Cheng, 2012). Line 6 finds all the $k$-exposed subgraphs by traversing over the edges via triangle connections, requiring $O(|\triangle(E)|)$ time, where $\triangle(E)$ is the list of all triangles in the graph. Complexity of Line 7 depends on the order-based algorithm by Zhang & Yu (2019), which is $O(|\triangle(TCE_S)|)$ time for the unit edge removal. Here, $|\triangle(TCE_S)|$ is the total time required to list all

Table 1: Statistics of the datasets and runtime results. #C is the number of classes, R is the ratio of smallest class size to the largest class size, denoting the class imbalance, and $\phi_m$ is the maximum trussness of a graph. $|\mathcal{S}_G|$ is the total number of $k$-exposed subgraphs and $|TCE|$ is the average number of truss changed edges after removing any edge $e \in G$. In the remaining columns, we compare the naive approach and $ERC$ algorithm, and show the fraction of edges processed by $ERC$ as well as its speedup (Sp.) against the naive approach. We also give the GNN runtime for edge classification and compare the $ERC$ runtime to the toal $ERC$+GNN computation.

| Graph | $|V|$ | $|E|$ | #edge attr. | #C | R | $\phi_m$ | $|TCE|$ | $|\mathcal{S}_G|$ | Naive runtime (seconds) | ERC runtime (seconds) | Frac. of edges proc. | Sp. | GNN runtime (seconds) | Frac. of ERC to total |
|---|---|---|---|---|---|---|---|---|---|---|---|---|---|---|
| AMiner | 40.0K | 105.4K | 256 | 10 | 0.1992 | 8 | 27 | 20.0K | 1.055 | 0.519 | 0.19 | 2.03 | 436.933 | 0.00119 |
| MAG | 40.0K | 120.4K | 256 | 10 | 0.0020 | 8 | 44 | 20.4K | 1.624 | 0.652 | 0.17 | 2.49 | 516.228 | 0.00126 |
| MIND | 242.9K | 2.1M | 256 | 10 | 0.3423 | 8 | 391 | 73.5K | 145.068 | 32.425 | 0.04 | 4.47 | 9322.486 | 0.00347 |
| BoT-IoT | 32.2K | 457.1K | 12 | 5 | 0.0041 | 4 | 3 | 156.7K | 4.080 | 1.932 | 0.34 | 2.12 | 1962.106 | 0.00098 |
| ToN-IoT | 38.9K | 124.9K | 12 | 10 | 0.0003 | 3 | 5 | 35.5K | 3.362 | 1.465 | 0.28 | 2.30 | 10612.520 | 0.00014 |
| UNSW-NB15 | 64.7K | 1.1M | 12 | 10 | 0.0001 | 5 | 12 | 469.5K | 414.707 | 45.894 | 0.44 | 9.04 | 20612.52 | 0.00222 |

the triangles containing $e$ for all $e \in TCE_S$. Hence, Line 7 will take $O(|\mathcal{S}_G| \cdot |\triangle(TCE_S)|)$ time where $\mathcal{S}_G$ stores all the $k$-exposed of graph $G$. Lines 8 to 13 have a time complexity of $O(|E| \cdot |TCE_S|)$, as we iterate over all the edges ($E$) and their corresponding $k$-exposed sets ($TCE_S$) in the loop. Hence, the overall time complexity is $O(|E|^{1.5} + |\mathcal{S}_G| \cdot |\triangle(TCE_S)| + |E| \cdot |TCE_S|)$. Note that, in practice $|\mathcal{S}_G|$ is notably smaller than $|E|$ and the average size of $TCE_S$ is also much smaller than $|E|$, as shown in Table 1.

**Space complexity.** In addition to the graph; $ts$ and $\phi$ require $O(|E|)$ space. In $TCE$ of Algorithm 1, we store the truss changed edges information for any $k$-exposed $S \in \mathcal{S}_G$, which takes $O(|\mathcal{S}_G| \cdot |TCE_S|)$ space. Finally, the $E'$ of the dependency graph takes $O(|E| \cdot |TCE_S|)$ to store $TCE_S$ for each edge. So, the overall space complexity is $O(|\mathcal{S}_G| \cdot |TCE_S| + |E| \cdot |TCE_S|)$.

## 5 EXPERIMENTAL EVALUATION

In this section, we evaluate our algorithms on several real-world datasets. We first investigate the distribution of the three new edge-based measures. Then we check the efficiency of **ERC** against the naive baseline. Next, we use our edge-based measures for the edge classification task and give a comprehensive evaluation on effectiveness and efficiency. All experiments are performed on a Linux operating system (v. 3.10.0-1127) running on a machine with Intel(R) Xeon(R) Gold 6130 CPU processor at 2.10 GHz with 192 GB memory. We implemented our **ERC** algorithm in C++. For the edge classification, we utilized the code from Wang et al. (2023), which is implemented in Python 3.9. **Our code and datasets are publicly available at `https://anonymous.4open.science/r/robustness/`.**

We consider six datasets for evaluation. Table 1 gives the details. AMiner (Tang et al., 2008) and MAG (Sinha et al., 2015) are co-authorship networks where nodes are scholars, edges represent couathorships, and edge classes are the papers' field of study. MIND (Wu et al., 2020) is a user interaction network from Microsoft News where edges connect users who clicked/viewed the same news and classes denote the news categories. BoT-IoT, ToN-IoT, and UNSW-NB15 (Sarhan et al., 2021) are constructed from NetFlow-based datasets, where nodes are port addresses and edges are interactions between ports. Edge features include network flow statistics and the class of edges are attack types.

**Truss robustness measures.** To understand whether the truss robustness measures, $ES$, $ER$, $EdgeRank$, capture something different than the triangle counts and trussness values, we compare their distribution. Figure 3 gives the results for AMiner graph (results for MAG is given in Figure 6 in Appendix). Feature values are shown on the $x$-axis, and corresponding frequencies are shown on the $y$-axis. $ES$, $ER$, and $EdgeRank$ all exhibit distinct distributions from each other and also show different behaviors than the trussness and triangle counts. This suggests that our edge-based robustness measures can be useful at discriminating between different types of edges. $ES$ and $ER$ typically exhibit left-skewed distributions whereas $EdgeRank$ has a bimodal behavior. The more skewed and spread-out distributions of $ES$ and $ER$ suggests wider variations in these aspects

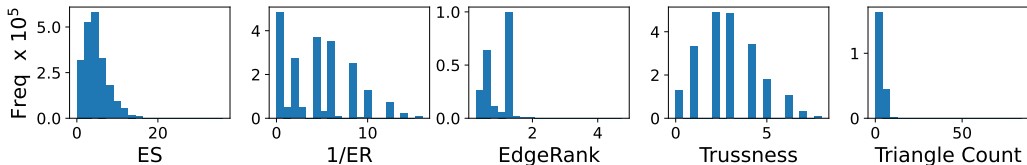

Figure 3: Value distribution of five different edge features—$ES$, $ER$, $EdgeRank$, trussness, and triangle count—for AMiner graph. $x$-axis represents the feature values and $y$-axis represents the corresponding frequency ($*10^5$).

Table 2: Macro F1 scores for edge classification on Poisson variant. Best scores are marked in bold for each graph and method. On average, merging ES, ER, and EdgeRank gives the best results.

| Graph | TER+AER | TER+AER +$RS\_OD$ +$RS\_ID$ | TER+AER +Trussness +Triangle c. | TER+AER +Coreness +Degree | TER+AER +$ES$+$ER$+ $EdgeRank$ |
|---|---|---|---|---|---|
| AMiner | 87.51 ± 1.49 | 87.12 ± 5.32 | 87.25 ± 2.55 | 87.17 ± 2.70 | **88.24 ± 0.75** |
| MAG | 85.71 ± 3.44 | 85.86 ± 2.97 | 87.06 ± 2.22 | 86.26 ± 2.08 | **88.39 ± 2.70** |
| MIND | 90.13 ± 0.07 | 92.50 ± 0.07 | 90.52 ± 0.09 | **92.70 ± 0.06** | 90.44 ± 0.07 |
| BoT-IoT | 50.27 ± 2.34 | 51.23 ± 3.01 | 51.38 ± 2.86 | 51.13 ± 2.63 | **53.1 ± 1.81** |
| ToN-IoT | 28.04 ± 0.52 | 28.4 ± 0.51 | 28.18 ± 0.29 | 28.16 ± 0.42 | **30.48 ± 0.64** |
| UNSW-NB15 | 11.64 ± 0.21 | 11.62 ± 0.12 | 11.55 ± 0.13 | 11.69 ± 0.15 | **14.72 ± 0.14** |

across edges, which is in line with the standard deviation numbers in Figure 2b. This indicates that they could be more useful for identifying edges with distinct local network structures. Besides, the bimodal distribution of $EdgeRank$ might be useful in distinguishing certain edge classes.

**Efficiency of ERC algorithm.** Here we compare the runtime performances of our **ERC** algorithm against the naive strategy which simply runs the algorithm designed by Zhang & Yu (2019) for each edge removal. Table 1 gives the absolute runtimes, fraction of edges processed by **ERC** against the baseline, and the overall speedup of **ERC**. Using our **ERC** algorithm, we process as low as 4% of edges compared to the naive method. On average, **ERC** is 3.74× faster than the baseline across all datasets and gives up to 9.04× speedup.

## 5.1 EDGE CLASSIFICATION

We consider the state-of-the-art edge classification method, TER+AER (both Poisson and Geometric variants), from (Wang et al., 2023) as the baseline and integrate our truss robustness measures into it for comparison. As an alternative baseline that uses higher-order information, we integrate triangle count and trussness values of edges into TER+AER. And as an alternative robustness baseline, we use the core robustness-based edge features from (Hossain et al., 2023), by aggregating the endpoint features of $RS_{OD}$, $RS_{ID}$, degree, and core number, and again integrate into TER+AER. We consider all the edge attributes given in the datasets, as in (Wang et al., 2023), and also keep the same parameters such as the GNN layer, dropout rate, Adam optimizer, and learning rate. Considering the significant imbalance in the datasets, particularly in MAG and NetFlow-based ones (see R in Table 1), we give the macro F1 scores for all approaches to better highlight the gains in minority classes. We conduct ten runs with different random seeds for each method on each graph and compute the averages and standard deviations.

Table 2 gives the results for Poisson variant of TER+AER (AUC results and Geometric variant of TER+AER are given in Section 7.5 at Appendix). Overall, integrating our measures provides consistently better performance than the vanilla TER+AER approach and other baselines. Although the integration of coreness and degree gives the best results on MIND, our approach outperforms the TER+AER baselines. In MIND, approximately 75% of edges have consistently high triangle counts (see Figure 7 at Appendix), which puts most of them in small $k$-exposed clusters, and hence they get the same $ER$ score, which hinders our performance. From similar experiments shown in Figure 2b, we observe that the structures contributing to the improved performance are similar to those in the toy graph in Figure 1 (details are in Figure 8 at Appendix). Besides, our hypothesis

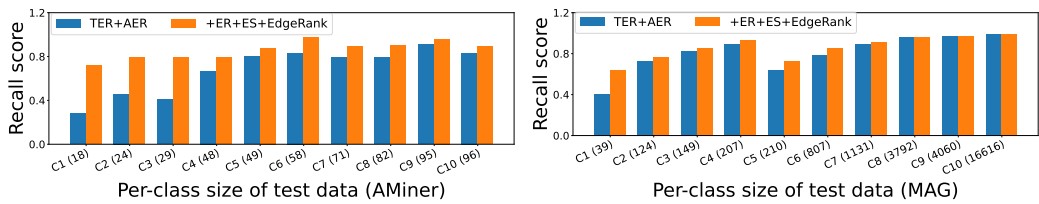

Figure 4: Per-class recall scores for edge classification task. $x$-axis is the class size for each edge class type and $y$-axis is the recall score of TER+AER and truss robustness measures.

that the higher standard deviations of class-wise average values ($ES$ and $EdgeRank$) in AMiner and MAG graphs would translate to improved performance on edge classification, as depicted in Figure 2b, is materialized in our experimental results.

We further evaluate our prediction performance on specific edge classes in AMiner and MAG graphs in Figure 4 (a more detailed result for AMiner is given in Section 7.6.3 at Appendix). We show the Recall scores (on the $y$-axis) for the vanilla TER+AER (Geometric) method and our approach which merges edge-based truss robustness features to TER+AER (Geometric) (Precision scores are similar). $x$-axis is the class size for each edge class type, sorted in increasing order of the class size. Recall performance gets significantly better for the rare classes after incorporating our measures. Unlike Wang et al. (2023), who reported low F1 scores on the AMiner graph due to its imbalanced nature, our measures yield high F1 scores.

Additionally, to visualize the impact of integrating ES+ER+EdgeRank into TER+AER on learned edge embeddings, we consider t-SNE plots focusing on three rarest classes, plots are given in Figure 10 at Appendix. We reduce the dimensionality of these embeddings to two components to effectively compare their distributions before and after integration. The visual difference between the plot pairs indicates that the inclusion of ES+ER+EdgeRank helps in distinguishing the edges more effectively.

**Ablation study.** To evaluate the contribution of each robustness measure, we conduct an ablation study. Each of the three robustness features improves the edge classification independently and using all collectively gives the best results consistently. Details are given in Section 7.6.4 at Appendix.

**GNN runtime comparison.** Lastly, we investigate the computational overhead of computing the edge-based truss robustness measures for the edge classification task. We show the runtime of the GNN in Wang et al. (2023), **ERC**, and the fraction of **ERC** to GNN+**ERC** as the average of ten runs in Table 1. For all the datasets, **ERC** takes significantly less time than the GNN. The computation time of **ERC** is negligible when compared to the GNN runtime.

## 6 CONCLUSION

In this paper, we characterized and utilized the robustness of truss decomposition in an edge-driven way. We proposed to build a dependency graph among edges to capture the impact on neighbor edges and devised three new edge-based truss robustness measures on it. We introduced several theoretical findings and designed an efficient algorithm to practically compute those measures. We showed the utility and practicality of our measures in the edge classification task for real-world networks.

**Limitations and future work.** While our work focuses on edge robustness under removal, exploring its behavior upon insertion presents a promising future direction. As pointed in Zhang & Yu (2019), updating the truss numbers upon insertions is a much more challenging problem with no known non-trivial upper bound. This is a limitation of the current approach and warrants further investigation. Moreover, it will be interesting to incorporate our measures into other graph analysis tasks, such as link prediction, and investigate their performances. Additionally, we plan to perform a theoretical analysis on synthetic graphs (with simple formation processes) to prove how truss robustness measures have a better discriminative power (than the other traditional graph analytics). Lastly, to assess node importance within the dependency graph, we employed efficient local (in-degree, out-degree) and global (PageRank) measures. More complex metrics, such as Eigenvector and Betweenness centrality, are computationally costly for our large dependency graphs, which we aim to explore in future work.

**Reproducibility Statement:** All of our experimental results are reproducible. The anonymous code is publicly available at `https://anonymous.4open.science/r/robustness/`, with more details provided in Section 5. The datasets we used are accessible at `https://mega.nz/folder/hj8EHIRR#ZAtmgX8eVao-FxScFqxHIQ`, with additional information available in the aforementioned code repository. Additionally, we include the Lemmas, their proofs, and the algorithm in Section 4 and Appendix (Section 7.3).

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

# 7 APPENDIX

## 7.1 NOTATION TABLE

Table 3: Notations.

| Notation | Description |
|----------|-------------|
| $\phi(e, G)$ | truss number (trussness) of edge $e$ in $G$ |
| $\phi_m$ | maximum truss number in the graph |
| $E(e, G)$ | set of incident edges to edge $e$ in $G$, i.e., share a triangle |
| $\Gamma_<(e, G)$ | incident edges of $e$ with smaller trussness |
| $sup(e, G)$ | support of an edge $e$ in $G$ |
| $ts(e, G)$ | support of an edge $e$ in $\phi(e)$-truss |
| $\triangle(e)$ | set of triangles that contain edge $e$ |
| $\triangle(E)$ | union of all $\triangle(e)$ for each edge $e \in E$ |
| $\mathcal{S}_G$ | set of all $k$-exposed in $G$ |

## 7.2 IMPLICATIONS OF TRUSS ROBUSTNESS MEASURES FOR EDGE CLASSIFICATION

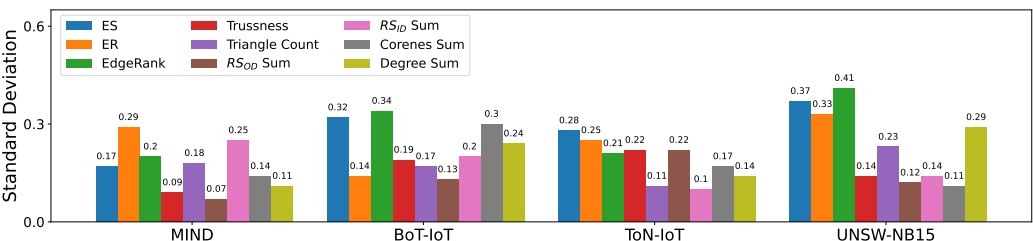

Figure 5: Standard deviation of class-wise mean values for each edge feature in four different datasets. The higher the standard deviation, the more distinguishing the edge feature is expected to be in the edge classification task. Overall, the higher standard deviation of $ER$, $ES$, and $EdgeRank$ features highlights their potential.

## 7.3 OMITTED PROOFS

**Proof of Lemma 3:** Any two edges $e, e'$ in a $k$-exposed $S$ are either in the same triangle or connected via a series of adjacent triangles. For the first case, if they are in the same triangle, then they will be one of the sensitive incident edges of one another. Hence, for those edges, decreasing the truss number on one edge will decrease the trussness support of another edge resulting in a decrease in the truss number. Otherwise, for the second case, the proof follows from the fact that there are (consecutive) pairs of triangles that connect $e$ to $e'$ where the first case holds for each edge pair. We define two edges to be incident if they are in a same triangle. As $k$-exposed is a triangle-connected subgraph, consider a path of incident edges between $e$ and $e'$, $\{e_1, e_2, \cdots, e_{n-1}, e_n\}$, in $S$ where $e_1 = e$, $e_n = e'$, and $e_1$ is incident to only $\{e_2\}$, $e_2$ is incident to only $\{e_1, e_3\}$, and $e_i$ is incident to only $\{e_{i-1}, e_{i+1}\}$ for $i < n$. Here, $e_1$ and $e_2$ are the sensitive incident edges of one another and from the first case, a decrease in $\phi(e_1)$ will also affect the $\phi(e_2)$. Likewise, edges $e_i$ and $e_{i+1}$ are also dependent on each other from the first case. As $e_1$ and $e_n$ are connected by a series of adjacent triangles, we can conclude that any two edges in a $S$ are dependent on each other, and a decrease in one edge's truss number will affect all other edge's truss numbers. As a result, truss number of all the edges $e \in S$ will change if we choose any edge's truss number to decrease, and the lemma holds.

**Proof of Lemma 4:** From Lemma 3, for any $k$-exposed $S$, all the edge's truss numbers will decrease if there is a decrease of truss number in any edge. Changes in the truss number of those edges may affect the truss number of some other edges $e' \in ST_e$ where $e \in S$ according to Lemma 2. Here, a decrease in the truss number of any edge is affecting the same set of edges in $S$; thus the decreases of

trussness support of their incident edges will be the same. As a result, a decrease in $\phi(e)$ for any edge $e \in S$ will affect all the edges $\forall e' \in S$ and their effect on Truss Changed Edges will be the same.

### 7.4 TCE COMPUTE ALGORITHM

---

**Algorithm 2: TCECompute** $(G(V, E), S, \phi, ts)$

1 **Input:** $G$ $(V, E)$: graph, $S$: $k$-exposed, $\phi$: trussness vector of $G$, $ts$: trussness support vector of $G$

2 **Output:** $TCE_S$: Truss changed edges of $S$

3 $Q \leftarrow []$           // empty queue to store truss changed edges

4 $e_{rnd} \leftarrow$ any random edge in $S$ to decrease it's $\phi(e_{rnd})$

5 $ts(e_{rnd}) \leftarrow ts(e_{rnd}) - 1$

6 $Q$.push($e_{rnd}$), $TCE_S$.push($e_{rnd}$)

7 **while** $Q$ *is not empty* **do**

8      $e_{dec} \leftarrow Q$.pop()

9      **foreach** *triangle* $\triangle \in \triangle(e_{dec})$ *in* $G$ **do**

10          let $e'$ and $e''$ be the other two edges of $\triangle$

11          update $ts(e')$ and $ts(e'')$ by inspecting $\phi(e_{dec}), \phi(e')$ and $\phi(e'')$

12      **foreach** $e \in \Gamma_{\leq}(e_{dec}, G)$ **do**

13          **if** $ts(e) < \phi(e)$ **then** $Q$.push($e$), $TCE_S$.push($e$)

14 Return $TCE_S$

---

#### 7.4.1 VALUE DISTRIBUTION OF MAG GRAPH

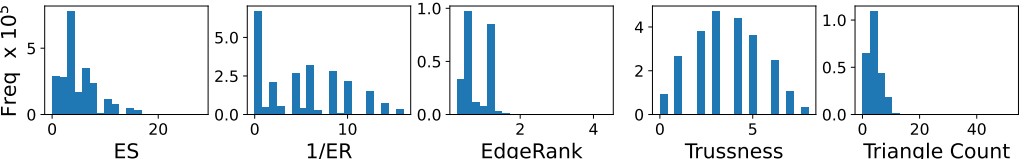

Figure 6: Value distribution of five different ($ES$, $ER$, $EdgeRank$, Trussness, and Triangle count) edge features of MAG graph. $x$-axis represents the feature values and $y$-axis represents corresponding frequency ($*10^5$).

### 7.5 EDGE CLASSIFICATION RESULTS

Table 4: Macro F1 scores for edge classification on Geometric variant. Best scores are marked in bold for each graph and each method. On average, merging edge-based truss robustness measures performs better than others.

| Graph | TER+AER | TER+AER +*RS_OD* +*RS_ID* | TER+AER +Trussness +Triangle c. | TER+AER +Coreness +Degree | TER+AER +*ES*+*ER*+ *EdgeRank* |
|---|---|---|---|---|---|
| AMiner | 87.56 ± 4.12 | 88.68 ± 3.58 | 86.39 ± 4.60 | 88.30 ± 3.40 | **89.59 ± 1.31** |
| MAG | 86.14 ± 1.70 | 86.57 ± 4.20 | 86.99 ± 3.47 | 86.58 ± 3.44 | **87.90 ± 1.82** |
| MIND | 92.76 ± 0.06 | 94.24 ± 0.05 | 93.08 ± 0.05 | **94.49 ± 0.08** | 92.99 ± 0.07 |
| BoT-IoT | 50.27 ± 2.34 | 51.23 ± 3.01 | 51.38 ± 2.86 | 51.13 ± 2.63 | **53.1 ± 1.81** |
| ToN-IoT | 28.04 ± 0.52 | 28.4 ± 0.51 | 28.18 ± 0.29 | 28.16 ± 0.42 | **30.48 ± 0.64** |
| UNSW-NB15 | 11.64 ± 0.21 | 11.62 ± 0.12 | 11.55 ± 0.13 | 11.69 ± 0.15 | **14.72 ± 0.14** |

Table 5: AUC scores for edge classification. G and P denote Geometric and Poisson variants of TER+AER. Best scores are marked in bold for each graph and each method. On average, merging edge-based truss robustness measures performs better than others.

| Graph | | TER+AER | TER+AER +RS_OD +RS_ID | TER+AER +Trussness +Triangle c. | TER+AER +Coreness +Degree | TER+AER +ES+ER+ EdgeRank |
|---|---|---|---|---|---|---|
| AMiner | G | 97.33 ± 0.3 | 97.39 ± 0.16 | 97.45 ± 0.28 | 97.46 ± 0.27 | **97.54 ± 0.31** |
| | P | 97.84 ± 0.2 | 97.8 ± 0.22 | 97.79 ± 0.2 | 97.85 ± 0.18 | **97.86 ± 0.2** |
| MAG | G | 99.26 ± 0.12 | 99.22 ± 0.1 | 99.32 ± 0.11 | 99.3 ± 0.1 | **99.34 ± 0.08** |
| | P | 98.76 ± 0.02 | 99.17 ± 0.01 | 98.94 ± 0.01 | **99.27 ± 0.02** | 99.26 ± 0.01 |
| MIND | G | 99.32 ± 0.05 | 99.3 ± 0.13 | 99.36 ± 0.1 | **99.37 ± 0.1** | 99.34 ± 0.04 |
| | P | 97.93 ± 0.02 | 98.71 ± 0.02 | 98.16 ± 0.01 | **98.83 ± 0.01** | 98.1 ± 0.02 |
| BoT-IoT | G | 93.93 ± 0.14 | 93.95 ± 0.23 | 93.87 ± 0.29 | 93.98 ± 0.25 | **94.12 ± 0.23** |
| | P | 94.08 ± 0.14 | 94.05 ± 0.18 | 94.1 ± 0.13 | 94.1 ± 0.11 | **94.23 ± 0.21** |
| ToN-IoT | G | 89.15 ± 0.08 | 89.11 ± 0.1 | 89.06 ± 0.1 | 89.19 ± 0.11 | **89.81 ± 0.09** |
| | P | 88.72 ± 0.06 | 88.71 ± 0.07 | 88.72 ± 0.07 | 88.75 ± 0.05 | **89.3 ± 0.05** |
| UNSW-NB15 | G | 87.09 ± 0.33 | 86.98 ± 0.4 | 86.91 ± 0.69 | 87.04 ± 0.43 | **88.65 ± 0.33** |
| | P | 86.96 ± 0.36 | 86.74 ± 0.37 | 86.89 ± 0.7 | 87.2 ± 0.34 | **88.23 ± 0.6** |

## 7.6 ADDITIONAL EXPERIMENTAL RESULTS

### 7.6.1 TRIANGLE COUNT DISTRIBUTION

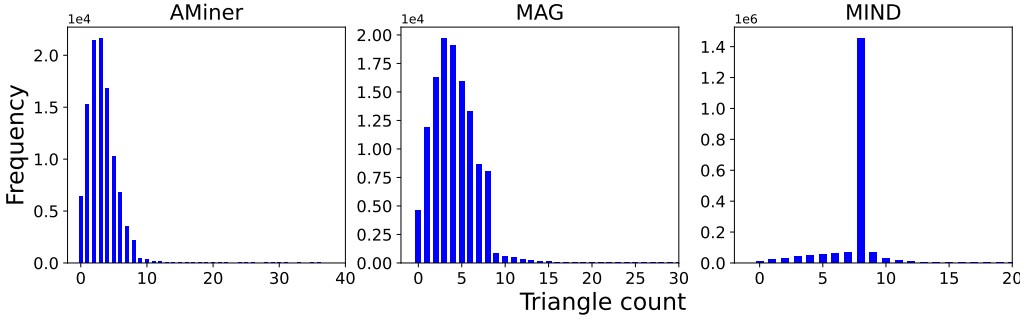

Figure 7: Triangle count distribution of three different datasets. $x$-axis represents the triangle count values and $y$-axis represents corresponding frequency

### 7.6.2 STRUCTURE CONTRIBUTING IMPROVED PERFORMANCE

When we check the edges solely identified by our edge robustness measures (but not by other methods), we notice that the structures that contribute to the improved performance are similar to the toy graph in Figure 1. The reasoning for this is similar to what we provided in Figure 2b. We focus on edges exclusively identified by our measures (and not by others). For each class, the average robustness of these edges are given in Figure 8. The standard deviation of these class-wise averages is high, which means our measure assigns distinct robustness values to different classes, unlike other measures whose stdevs remain low across classes. In simpler terms, while other measures (e.g., trussness, triangle count) tend to provide similar values across classes, ours provide different values across different classes (thanks to the structures in Figure 1). This ability of edge robustness to differentiate across classes is the key factor contributing to the superior performance.

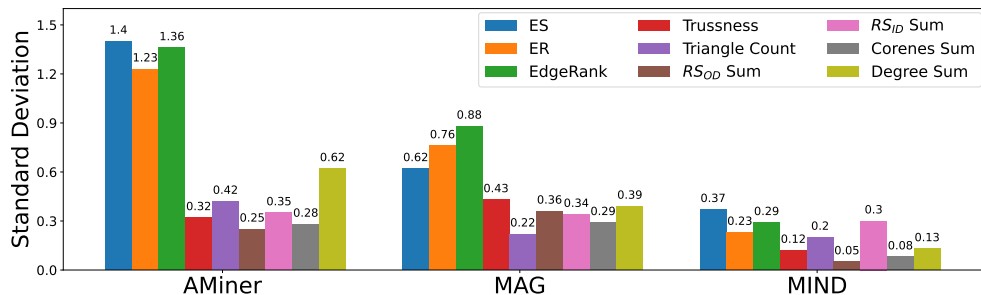

Figure 8: Standard deviation of class-wise mean values on edges exclusively identified by +ER+ES+EdgeRank for each edge feature. Higher standard deviations for ER, ES, and EdgeRank features indicate their potential for effective edge classification, as a larger standard deviation suggests a more distinguishing feature.

### 7.6.3 ALL FEATURES RECALL SCORE ON AMINER GRAPH

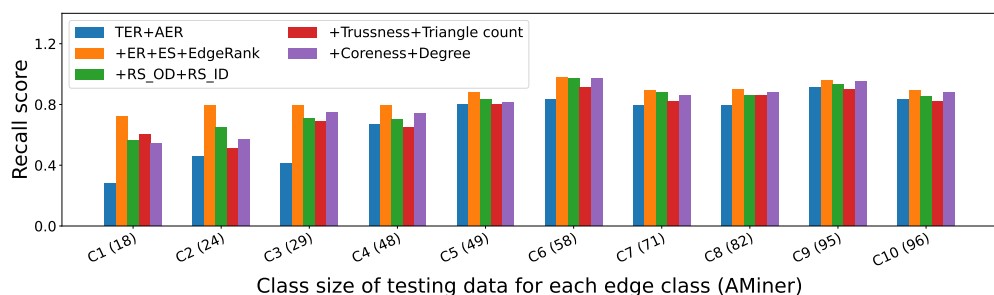

Figure 9: Per-class recall scores for edge classification task (AMiner). $x$-axis is the class size for each edge class type and $y$-axis is the recall score of TER+ AER and after integrating all other measures with TER+AER.

### 7.6.4 ABLATION STUDY RESULTS

Having demonstrated the overall performance boost from using all three truss robustness measures, we now conduct an ablation study to evaluate the individual contributions. We repeat each experiment ten times using different random seeds and consider the average results. The results are given in Table 7. Notably, excluding any single measure results in a performance decline, underlining the individual significance of each of the three measures. As suggested by Figure 2b, excluding $ES$ or $EdgeRank$ significantly reduces the performance for the MAG graph, whereas from Figure 5, omitting $ER$ affects the MIND graph performance. Importantly, combining all measures yields the highest performance in all cases. Also, the enhanced performance on AMiner and MAG graphs with $ES$ and $EdgeRank$ measures supports their collective contribution, as anticipated in Figure 2b.

Table 6: Edge classification performance (F1 scores) with the integration of all features. G and P represent the Geometric and Poisson variants of TER+AER. The best scores are highlighted in bold for each graph and method. The final column shows the results after merging all features, including TER + AER + RS_OD + RS_ID + Trussness + Triangle count + Coreness + Degree + ES + ER + EdgeRank.

| Graph | G/P | TER+AER | TER+AER +RS_OD +RS_ID | TER+AER +Trussness +Triangle c. | TER+AER +Coreness +Degree | TER+AER +ES+ER+ EdgeRank | TER+AER +All Features |
|---|---|---|---|---|---|---|---|
| AM | G | 87.56 ± 4.12 | 88.68 ± 3.58 | 86.39 ± 4.60 | 88.30 ± 3.40 | 89.59 ± 1.31 | **90.94 ± 1.31** |
| | P | 87.51 ± 1.49 | 87.12 ± 5.32 | 87.25 ± 2.55 | 87.17 ± 2.70 | 88.24 ± 0.75 | 89.18 ± 2.67 |
| MA | G | 86.14 ± 1.70 | 86.57 ± 4.20 | 86.99 ± 3.47 | 86.58 ± 3.44 | 87.90 ± 1.82 | 88.45 ± 2.78 |
| | P | 85.71 ± 3.44 | 85.86 ± 2.97 | 87.06 ± 2.22 | 86.26 ± 2.08 | 88.39 ± 2.70 | **89.23 ± 3.11** |
| MI | G | 92.76 ± 0.06 | 94.24 ± 0.05 | 93.08 ± 0.05 | 95.41 ± 0.03 | 92.99 ± 0.07 | **95.70 ± 0.06** |
| | P | 90.13 ± 0.07 | 92.50 ± 0.07 | 90.52 ± 0.09 | 92.70 ± 0.06 | 90.44 ± 0.07 | 92.95 ± 0.08 |

Table 7: Ablation study showing the performance contribution of each edge robustness measure (higher the better). G and P denote Geometric and Poisson variants of TER+AER. Best F1 scores are marked in bold across different combined features.

| Graph | | TER+AER | TER+AER + ES + ER | TER+AER + ES +EdgeRank | TER+AER + ER + EdgeRank | TER+AER + ES + ER + EdgeRank |
|---|---|---|---|---|---|---|
| AMiner | G | 87.56 ± 4.12 | 88.51 ± 3.17 | 89.07 ± 2.99 | 88.17 ± 3.69 | **89.59 ± 1.31** |
| | P | 87.51 ± 1.49 | 86.82 ± 2.55 | 87.51 ± 3.84 | 86.51 ± 2.77 | **88.24 ± 0.75** |
| MAG | G | 86.14 ± 1.70 | 85.37 ± 2.87 | 86.84 ± 2.03 | 85.89 ± 2.58 | **87.90 ± 1.82** |
| | P | 85.71 ± 3.44 | 85.18 ± 2.41 | 85.07 ± 2.63 | 85.82 ± 2.40 | **88.39 ± 2.70** |
| MIND | G | 92.76 ± 0.06 | 92.92 ± 0.08 | 92.92 ± 0.07 | 92.88 ± 0.07 | **92.99 ± 0.07** |
| | P | 90.13 ± 0.07 | 90.38 ± 0.05 | 90.39 ± 0.10 | 90.38 ± 0.08 | **90.44 ± 0.07** |

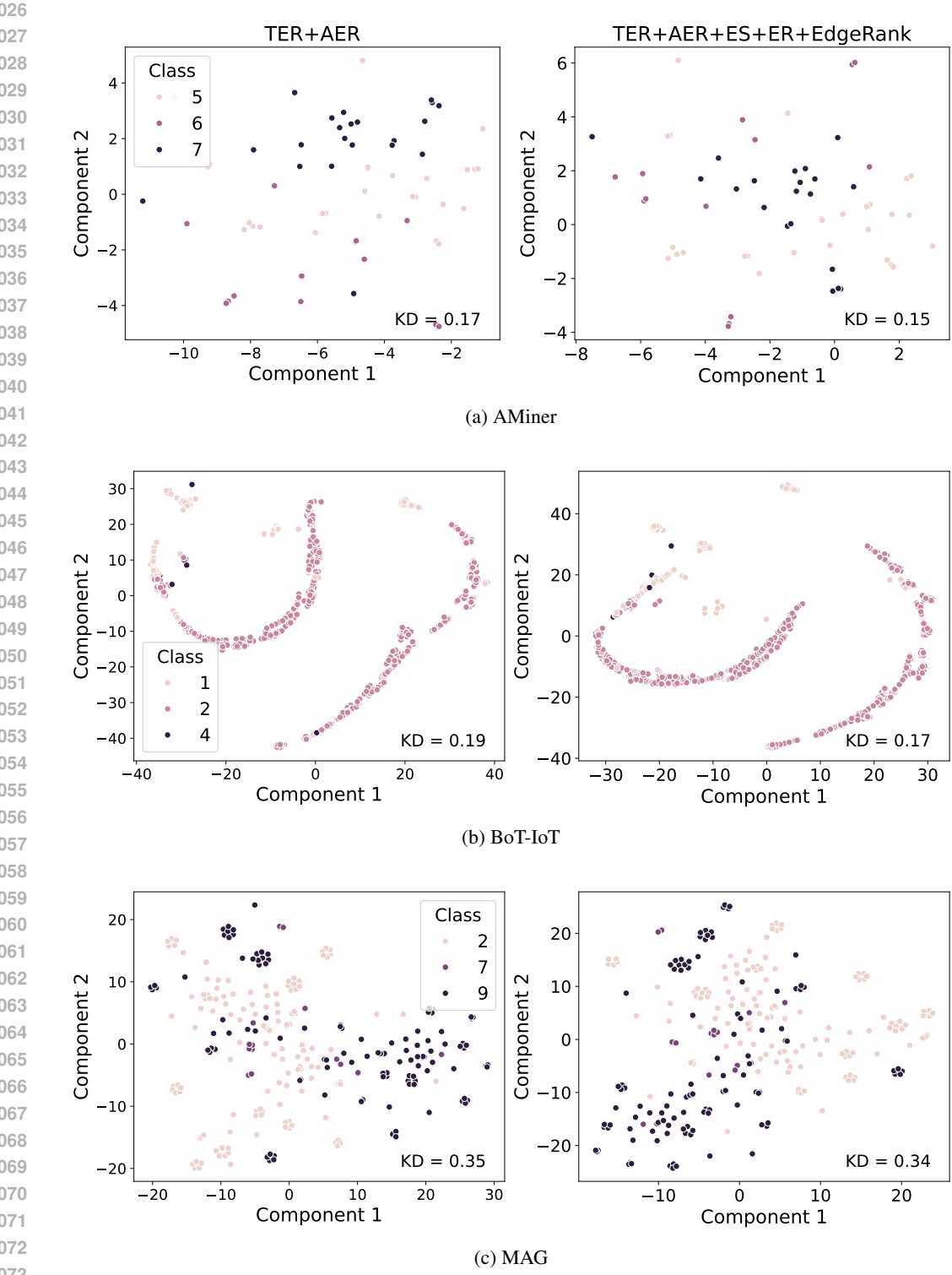

Figure 10: t-SNE plots illustrating the learned edge embeddings before (left) and after (right) integrating ES+ER+EdgeRank with TER+AER on three datasets: AMIner, BoT-IoT, and MAG. The Kullback-Leibler (KL) divergence (denoted as KD) represents the difference between the high-dimensional and low-dimensional probability distributions of the t-SNE projection. Lower KL divergence signifies a better representation of the data in the reduced dimensional space.

