# OpenReview forum: "Robustness of Truss Decomposition and Implications for GNN-based Edge Classification"
_ICLR.cc/2025/Conference — ICLR 2025 Conference Withdrawn Submission_

### Official Review · Reviewer_kpuv · 2024-10-30

**Soundness:** 2
**Presentation:** 1
**Contribution:** 2
**Rating:** 3
**Confidence:** 4

**Summary:**

This paper, titled "Robustness of Truss Decomposition and Implications for GNN-Based Edge Classification," addresses the sensitivity of truss decomposition in dense subgraph discovery. Truss decomposition is noted to be highly effective but sensitive to small changes, like edge removals, which significantly impact edge truss values. The authors propose a new framework for characterizing truss robustness on an edge level by constructing a dependency graph that captures the impact of each edge's removal on its neighbors. They further use the captured robustness and dependencies in downstream edge classification problem via GNN.

**Strengths:**

1. The abstract is well-written.
2. The observation of this paper is insightful.
3. The proposed method is interesting and mathematically grounded.

**Weaknesses:**

1. In section 2, the paper introduces several truss-related concepts (e.g., truss number and trussness support), which can initially be confusing, especially the distinction between trussness support and truss number. An example would help clarify these concepts and highlight their differences. Additionally, the definition of trussness support in the formula (line 137) is missing the cardinality notation "∣∣" and should be corrected for clarity.
2. In Figure 2(a), the dependency graph does not fully align with the truss number definition. For example, there should be a single directed edge between e2 and e1, and an edge should also exist between e2 and e5, right? Furthermore, the statement “as is the case (e3, e5) for which are incident on the left but not connected on the right” contradicts figure 2(a), as e3 and e5 are indeed unconnected in the dependency graph. 3.
3. In Section 3, the paper lacks a formula for computing EdgeRank, which reduces the transparency and reproducibility of the method.
4. In Figure 2(b), Edge Robustness (ER) shows a relatively low standard deviation, yet no explanation is provided. It would be helpful to discuss why ER might show limited variability across classes.
5. The Experiments section lacks a direct comparison with the baseline from Chen et al. (2021) and omits runtime data for other robustness indicators like RS_{OD}, RS_{ID}, degree, and core number, which makes the efficiency claims not fully supported by experimental results. Adding a comparison with Chen et al. (2021) and reporting the runtime of other measures would provide a more comprehensive evaluation of computational efficiency and better support the paper’s claims.
6. According to Table 5, the improvement of the proposed method over coreness+degree is quite marginal, and perhaps coreness is easier to compute than the metrics proposed in this paper. Any justifications or explanations?
7. Are there other combinations (among metrics proposed in this paper and previous degree, coreness, etc) that could achieve better results? Seems they can be combined?

Huiping Chen, Alessio Conte, Roberto Grossi, Grigorios Loukides, Solon P Pissis, and Michelle Sweering. On breaking truss-based communities. In Proceedings of the 27th ACM SIGKDD Conference on Knowledge Discovery & Data Mining, pp. 117–126, 2021.

**Questions:**

Please refer to the weakness

---

> ### Author Response · Authors · 2024-11-25
>
> Thank you for your comments. Here, we address the Weaknesses you mentioned.
>
> **W1**: We did not introduce truss-related concepts such as truss number and trussness support; these terms were defined in earlier works (Cohen, 2008; Zhang & Yu, 2019). We have used these terms as described in those papers and provided their definitions in Lines 128-130 and Lines 135-136. Additionally, Figure 1 illustrates information related to the truss number and triangle count (which is related to trussness support). For further clarification, we recommend referring to the original papers. Additionally, thank you for pointing out the missing cardinality notation; we have updated it in the revised version.
>
> **W2**: Thank you for highlighting the inconsistency in the dependency graph. The descriptive text is accurate; however, the edge numbering (in 2 pairs) was assigned incorrectly. We have updated the figure in the revised PDF for clarity.
>
> **W3**: The detailed description of EdgeRank is provided in Lines 231-237. We have added the new formula and its description in Lines 238-241 of the revised PDF.
>
> **W4**: We acknowledge that ER shows a low standard deviation in Figure 2b. However, additional results provided in the Appendix (Figure 7) show that ER exhibits higher standard deviation on other graphs. While Figure 2b might suggest that ER contributes less, the ablation study results and Figures 7 and 8 demonstrate that ER also plays a significant role in improving edge classification performance.
>
> **W5**: Please note that Chen et al.'s approach is not directly comparable to ours. While their work focuses on addressing the community breaking problem to make a graph k-truss free, our study centers on edge-based truss robustness, making the two approaches fundamentally different. Chen et al. examine the robustness and stability of communities, whereas we focus on the robustness of individual edges. We provide runtime data to show that the ERC algorithm is more efficient than naive baselines. Other metrics (like RS_OD, RS_ID, degree, and core number) are included to show how effective our measures are compared to existing ones. Hence, we didn’t include the runtime of other metrics.
>
> **W6**: Although the improvement over coreness+degree in Table 5 may appear modest, our proposed metrics show an improvement of up to 3.03% (on the UNSW-NB15 graph). Additionally, we provide the p-values from the t-test comparing the last two columns (coreness+degree vs. our metrics) for each graph. These results confirm that the performance improvements in our metrics are statistically significant.
>
> | Graph   | p-value          |
> |--------------|------------------|
> | AMiner       | 0.04416100726    |
> | MAG          | 0.001182955794   |
> | MIND         | 1.18E-72         |
> | BoT-IoT      | 0.00139144425    |
> | ToN-IoT      | 5.34E-22         |
> | UNSW-NB15    | 4.13E-61         |
>
>
> **W7**: In this paper, we focused on evaluating the individual contributions of truss robustness metrics. We also conducted experiments combining all the features, which resulted in better overall performance. The results are provided in Table 6 at Appendix of the revised pdf. While merging all the features yields the best results, the most significant improvement comes from our truss robustness metrics, as supported by the t-test p-values provided above in W6.

---

> > ### Comment · Reviewer_kpuv · 2024-11-28
> >
> > Thanks for your response. However, I'm not fully convinced, especially for W5 and W6. Why not report runtime for other metrics? W6 What's the effect side (e.g., Cohen’s d) of your improvement apart from p-values?

---

> > > ### Author Response · Authors · 2024-11-29
> > >
> > > Thank you for your response and concerns. Please find our responses to your concerns on W5 and W6 below.
> > >
> > > **Runtime for other metrics (W5):**
> > > We have conducted new experiments to obtain the baseline runtime results. Before discussing the results, let us first mention that the degree, core number, trussness, and triangle count are simpler graph properties. Computation of degree and core number has a time complexity of O(|E|), while trussness and triangle count have complexities of O(|E|^1.5) and O(|△(E)|), respectively (where △(E) is the list of all triangles). RS_{OD}, RS_{ID}, and our truss robustness metrics are built on these simpler properties, with some additional computational overhead. Computing RS_{OD} and RS_{ID} (given the core numbers) takes O(|V| $\cdot$ |E|) time. Our truss robustness metrics rely on trussness and triangle count, with the detailed complexity analysis provided in Lines 374-377 and 394-397.
> > > Runtime results are as follows (in seconds):
> > >
> > > | Graph          | Coreness + Degree | RS_OD + RS_ID | Trussness + Triangle Count | ERC Algorithm | ERC Speedup vs RS_OD + RS_ID |
> > > |----------------|-------------------|--------------|----------------------------|---------------|--------------------------------|
> > > | AMiner         | 0.62              | 35.69        | 0.66                       | 0.52          | 68.80                          |
> > > | MAG            | 0.61              | 21.95        | 0.69                       | 0.65          | 33.67                          |
> > > | MIND           | 6.70              | 579.87       | 9.34                       | 32.42         | 17.88                          |
> > > | NF-BoT-IoT     | 2.07              | 159.32       | 2.23                       | 1.93          | 82.45                          |
> > > | NF-ToN-IoT     | 1.55              | 23.10        | 1.70                       | 1.46          | 15.77                          |
> > > | NF-UNSW-NB15   | 17.92             | 177.36       | 22.84                      | 45.89         | 3.86                           |
> > >
> > > As expected the coreness, degree, trussness, and triangle count metrics are computationally efficient due to their simpler calculations. In comparison, the computation of RS_{OD} and RS_{ID} takes significantly longer, with coreness + degree being approximately 70 times faster. On the other hand, our algorithm has a comparable performance with trussness + triangle count (as well as coreness, degree). Additionally, our ERC algorithm is faster than the computation of RS_{OD} and RS_{ID}, primarily due to the reduced number of edge removals in our approach. This results in a significant runtime efficiency, with our approach being about 37 times faster. These results emphasize the efficiency of our approach compared to the RS_{OD} and RS_{ID} (while its effectiveness is already confirmed by the results in Table 2). We will include these additional results in the Appendix.
> > >
> > > **Cohen’s d of p-values (W6):**
> > > Here, we present an additional column for Cohen's d as requested by the reviewer.
> > >
> > > | Graph         | p-value          | Cohen's d  |
> > > |---------------|------------------|------------|
> > > | AMiner        | 0.04416100726    | 0.540      |
> > > | MAG           | 0.001182955794   | 0.884      |
> > > | MIND          | 1.18E-72         | -34.667    |
> > > | BoT-IoT       | 0.00139144425    | 0.873      |
> > > | ToN-IoT       | 5.34E-22         | 4.286      |
> > > | UNSW-NB15     | 4.13E-61         | 20.884     |
> > >
> > > The results suggest that our metrics have meaningful improvements across different datasets. For instance, AMiner exhibits a medium effect size of 0.54, indicating a meaningful performance difference. MAG and BoT-IoT show large effect sizes of 0.88 and 0.87, respectively, highlighting substantial improvements. ToN-IoT and UNSW-NB15 demonstrate extremely large effect sizes, with values of 4.29 and 20.88, indicating overwhelming improvements. These results suggest that our approach is not just statistically significant but also practically impactful across several datasets.
> > >
> > > We believe these responses have addressed all of the reviewers' concerns. If there is anything else we can clarify or provide to improve the score, please let us know. We would be glad to provide further details.

---

> > > > ### Author Response · Authors · 2024-11-30
> > > >
> > > > Thanks again for your valuable feedback. Could you please acknowledge that you read our response to your comments? If your concerns are addressed, we’d be grateful if you can adjust the score. If not, we’d love to engage further to address your comments.

---

> > > > ### Comment · Reviewer_kpuv · 2024-12-02
> > > >
> > > > Thanks for the authors' prompt and detailed response. However, the new experimental results open more questions. Regarding W6, why does Cohen's d give a negative value on MIND?
> > > > Regarding W5, as shown in the new results, ERC is lower than Coreness + Degree and has worse accuracy. It seems the motivation to use this new model and algorithm is not very strong. Hence, I'd like to retain my score.

---

> > > > > ### Author Response · Authors · 2024-12-02
> > > > >
> > > > > Thank you for your response. Among the six graphs, only in the MIND graph does Coreness + Degree achieve higher accuracy than our proposed metrics (see Table 2). As a result, Cohen's d yields a negative value for MIND. For the runtime comparison with Coreness + Degree, we would like to mention that our approach demonstrates comparable performance, with better runtime observed in half of the datasets.

---

### Official Review · Reviewer_oeMQ · 2024-11-02

**Soundness:** 3
**Presentation:** 2
**Contribution:** 2
**Rating:** 5
**Confidence:** 4

**Summary:**

This paper aims to study the edge level truss robustness and improve the performance of edge classification.
The authors propose three metrics to measure the truss robustness based on the dependency graph.
To speed up the computation, they propose an algorithm based on the theorems of truss number computation.
The experiments of edge classification have been conducted on six real-world graphs.

**Strengths:**

1. Theoretical analysis is provided.
2. The algorithm for fast computation is proposed based on theorems.

**Weaknesses:**

1. The writing of the paper can be improved.
2. Crucial evaluations of the proposed metrics are missing.

**Questions:**

1. In line 96, use the math symbol “$\times$” instead of an English character “x”.
2. In line 104, the word “cutting-edge” is overly strong.
3. In Figure 2b, why use standard deviation (STD) to measure the importance of edge features? First, are all the features normalized to ensure their STDs are comparable? Second, if the goal is effective classification, why not use the idea of linear probing and report the classification performance of a linear classifier?
4. In Section 3, last paragraph, continuing from the previous question, the role of this paragraph is unclear to me. These metrics are proposed to measure the truss robustness of an edge. Instead of showing how precise these metrics measure the truss robustness, this paragraph shows they are useful for edge classification. A paragraph showing how well these metrics measure robustness should be provided.
5. In Section 4, what is the time complexity of the naive computation of the dependency graph? How much faster is the proposed algorithm?
6. In Figure 3, the same problem as Question 4, showing the proposed metrics have different distributions from the existing ones does not justify their correctness. The important thing is to measure how accurate these metrics are in estimating truss robustness.
7. In Table 2, the last two columns seem to be statistically tied. Could you provide the p-values from the t-test?
8. In summary, in my opinion, it is important to study the truss robustness of the edges and have a fast algorithm. However, the writing of this paper and the title seem to emphasize its usefulness on edge classification. While the first part lacks crucial evaluations and data analysis, the second part lacks novelty. It would be helpful if the authors can clarify this.

---

> ### Author Response · Authors · 2024-11-25
>
> Thank you for your comments. First, we address the Weaknesses you mentioned.
>
> **W1**: We have addressed all of the questions you raised. We would be happy to address any specific comments you have on improving the writing of the paper.
>
> **W2**: Please see our answers to your questions below.
>
>
> Now, we would like to address the question you asked,
>
> **Q1**: Thank you for pointing this out. We have updated the symbol in the revised pdf.
>
> **Q2**: Wang et al. (2023) proposed the TER+AER approaches, which are state-of-the-art models for edge classification. Thus, we referred to their GNN-based approach as "cutting-edge," but it can be replaced with "recent" if preferred.
>
> **Q3**: The standard deviation is used as a preliminary analysis to examine the variability of edge feature values across different classes, helping identify features that may contribute to distinguishing edges.
>
> We have normalized all features, as mentioned in Line 256.
>
> We appreciate the suggestion of using linear probing for assessing classification effectiveness. However, our goal was to present the importance of edge robustness in a simpler and more statistical manner, rather than employing machine learning approaches. Besides, our truss robustness measures capture complex relationships in graphs that linear probing cannot easily evaluate. The connections between edges in truss structures are non-linear, making them harder for linear probing to handle. Additionally, our datasets are imbalanced, making linear probing less effective at identifying rare classes.
>
>
> **Q4**: The proposed metrics inherently measure truss robustness. For instance, Edge Robustness quantifies an edge's ability to retain its truss number when a neighboring edge is removed, while Edge Strength measures an edge's influence on changing the truss number of other edges. Note that the dependency graph is constructed by removing each edge in the graph (which is optimized in our algorithm), and the robustness metrics are derived from this process. Thus, these metrics inherently capture truss robustness without requiring additional validation. Additionally, the last paragraph highlights the practical value of these metrics by demonstrating their usefulness in tasks like edge classification.
>
> **Q5**: The runtime of our ERC algorithm is $O(|E|^{1.5} + |\mathcal{S}_G| \cdot |\triangle(TCE_S)| + |E| \cdot |TCE_S|)$, as stated in Line 397. In a naive approach, all edges in the graph would need to be removed instead of just the k-exposed edges. This would result in a runtime of $O(|E|^{1.5} + |E| \cdot |\triangle(TCE_S)| + |E| \cdot |TCE_S|)$.
>
> On average, our algorithm is 3.74 times faster than the naive baseline. This is mentioned in Line 462, and detailed results can be found in columns 10, 11, and 13 of Table 1
>
>
> **Q6**: The purpose of Figure 3 is to highlight how the proposed metrics differ from existing ones, supporting their potential use in edge classification. Please refer to Lines 430-432 and Lines 455-457 for more details. For the relevance of our new metrics in measuring truss robustness, please see our response to Question 4 above.
>
> **Q7**: The last two columns are NOT statistically tied. Below, we provide the p-values of t-test for each graph, which demonstrate that the performance improvements in our metrics are statistically significant.
>
>
> | Graph   | p-value          |
> |--------------|------------------|
> | AMiner       | 0.04416100726    |
> | MAG          | 0.001182955794   |
> | MIND         | 1.18E-72         |
> | BoT-IoT      | 0.00139144425    |
> | ToN-IoT      | 5.34E-22         |
> | UNSW-NB15    | 4.13E-61         |
>
>
>
> **Q8**: While the primary focus of this work is to study truss robustness and propose an efficient algorithm, we have also included edge classification as an application to demonstrate the practical utility of the metrics.
>
> To address the comment that “the first part lacks crucial evaluations and data analysis, and the second part lacks novelty”, we would like to clarify:
>
> In the first part of the paper, we introduced the truss robustness metrics and provided a thorough analysis of their usefulness. Specifically, we conducted a standard deviation analysis of two datasets in Figure 2b, demonstrating the value of truss robustness for edge classification. The results on three other datasets are provided in Figure 7 at Appendix. **Please let us know what other evaluation you’d like to see.**
>
> In the second part, we focused on applying these metrics to the edge classification task. The novelty of the paper is primarily in the introduction of the new feature, which has potential to contribute to the edge classification task. The per-class recall scores from Figure 4, show that truss robustness significantly enhances performance in rare classes and when dealing with imbalanced datasets (see Lines 501-508).
>
> As the proposed metrics offer new insights and improved performance in edge classification, we believe our work has novelty.

---

> > ### Comment · Reviewer_oeMQ · 2024-11-26
> >
> > I thank the authors for the responses. I have reviewed the responses and increased the score accordingly.

---

> > > ### Author Response · Authors · 2024-11-27
> > >
> > > We sincerely thank the reviewer for reviewing our responses and increasing the score.
> > >
> > > We have carefully considered and responded to all the points and concerns raised by the reviewer. If there is anything further we can clarify (or do) to improve the score, please let us know. We would be happy to provide additional details.

---

### Official Review · Reviewer_Ujfr · 2024-11-02

**Soundness:** 3
**Presentation:** 3
**Contribution:** 3
**Rating:** 8
**Confidence:** 3

**Summary:**

This paper introduces novel measures for edge-based robustness in truss decomposition, a method for dense subgraph discovery. The authors propose constructing a dependency graph among edges to model truss robustness and introduce three measures: Edge Robustness, Edge Strength, and EdgeRank. They provide theoretical findings and an efficient algorithm for computing the dependency graph. The paper demonstrates the effectiveness of these measures in improving edge classification tasks using Graph Neural Networks (GNNs).

**Strengths:**

1. The study presented in the paper fills a gap in the literature.
2. The toy exmple in Figure 1 is very helpful in understanding the concept.
3. The proposed measures show potential in improving downstream tasks like edge classification, particularly for rare classes in imbalanced datasets.

**Weaknesses:**

1. The paper primarily focuses on edge classification to demonstrate the effectiveness of the proposed measures. Exploring other applications could strengthen the work's impact.
2. Comparison with core decomposition SOTA measures could provide more context for the proposed measures' effectiveness.

**Questions:**

1. Have you considered applying these measures to other edge-centric tasks beyond classification, such as link prediction or graph matching?
2. How sensitive are the proposed measures to noise or small perturbations in the graph structure? Is there a way to quantify this sensitivity?

---

> ### Author Response · Authors · 2024-11-25
>
> Thank you for your comments. First, we address the Weaknesses you mentioned.
>
> **W1**: To demonstrate the usefulness of the proposed truss robustness metric, our study primarily focuses on edge classification. We agree that exploring other applications would better show its usefulness, and we have mentioned in our future work (see Lines 534-535) plans to apply this metric to tasks like link prediction.
>
>
> **W2**: We have already provided the performance of core decomposition metrics, including the core number sum. Its comparable performance is reported in Table 2 (5th column). Please refer to Lines 471-472 for more details.
>
>
> Now, we would like to address the questions you asked,
>
> **Q1**:  While our current work focuses on edge classification as an initial application of truss robustness, we acknowledge its potential for other edge-centric tasks, such as link prediction. We have already noted in our future work (see Lines 534-535) plans to extend the application of truss robustness to these tasks and explore its broader implications in graph representation learning.
>
> **Q2**: Thank you for pointing this out. Although the current study does not explicitly measure the sensitivity of our truss robustness metrics to noise, we intend to address this in future research. By conducting controlled perturbation experiments, we aim to gain a deeper understanding of how noise influences truss robustness scores.

---

> > ### Comment · Reviewer_Ujfr · 2024-11-27
> >
> > Understood. I would like to thank the authors for their detailed response.

---

> > > ### Author Response · Authors · 2024-11-28
> > >
> > > Thank you so much for your acknowledgement. We would be grateful if you could champion our work to further support its acceptance.

---

### Official Review · Reviewer_Hyhz · 2024-11-04

**Soundness:** 3
**Presentation:** 2
**Contribution:** 2
**Rating:** 5
**Confidence:** 3

**Summary:**

This paper quantifies the effect of removing an edge from a graph on the truss decomposition result. The authors construct a dependency graph to compute truss robustness of each edge and propose a faster heuristic based on their theoretical findings. The authors also show the effectiveness and efficiency of the proposed truss robustness to the edge classification task.

**Strengths:**

1.  The idea of truss robustness and dependency graph is intuitive and interesting.
2.  Theoretical findings in section 4 make the process of computing truss robustness efficient.

**Weaknesses:**

1. I like the first half of the paper, including the whole idea and conceptualisation of truss robustness and subsequent optimisation. However, it is unclear how truss robustness or truss decomposition effect on edge classification tasks. The authors present an interesting and computable quantitative metric for each edge, but where the metric can be effectively applied should be elaborated. It seems intuitive to me that there exists a significant portion of graph edge classifications that are not sensitive to truss robustness at all.
2. The applicability of truss robustness seems slightly narrower due to the fact that it can be used only as a feature for edge classification. Truss robustness is expected in the study of other edge-based tasks in graph representation learning such as link prediction.
3. Moreover, only one edge classification model TER-AER was reported in experimental result. And more experiments to verify the effectiveness of truss robustness on edge classification tasks are expected. Given the results so far, it seems that the entire work's only proposes a new feature for one model  on edge classification task, making it appear that the potential impact of the entire work is limited.

Assorted minor comments:

1.  I recommend that all mentioned notations should appear in Table 3.
2.  In time complexity analysis: $|E^{1.5}| \rightarrow |E|^{1.5}$
3.  I suggest that the authors use a different notation to indicate that the set of edges sharing a triangle with a particular edge  (i.e., $E(e, G)$) to distinguish from the notation of set consisting of all the edges of the graph.
4.  In Line 137, $ts(e,G)=\Gamma_{\geq}(e,\phi(e)\text{-truss})/2 \rightarrow ts(e,G)=|\Gamma_{\geq}(e,\phi(e)\text{-truss})|/2$ ?

**Questions:**

1. Can the authors go into more far-reaching detail about how truss robustness can help with the edge classification task?
2. Are there any other edge classification models apart from TER+AER for which truss robustness can be applied?
3. Are there any hyperparameters such as damping factor in edgerank? If so, how are these parameters chosen?

---

> ### Author Response · Authors · 2024-11-25
>
> Thank you for your comments. First, we address the Weaknesses you mentioned.
>
> **W1**: Truss robustness measures the structural importance and stability of edges within their local graph topology, capturing relationships often overlooked by traditional features. This makes it a valuable addition to edge classification tasks. In Section 3, we detail the implications of truss robustness for edge classification, highlighting its effectiveness (see Lines 241-262 and Figure 2b for more information). While we acknowledge that truss robustness may not be beneficial for every edge, it can effectively distinguish edges that other metrics fail to differentiate (as shown in Figure 1). Furthermore, the per-class recall scores provided in Figure 4 demonstrate that truss robustness improves performance, particularly in rare classes and when dealing with imbalanced datasets (see Lines 501-508).
>
> **W2**: We appreciate your concern regarding the applicability of truss robustness. As the first to introduce edge-based robustness, we used edge classification to illustrate its practical relevance and effectiveness. While our work centers on edge classification, we recognize the potential applications of truss robustness in other edge-based tasks, such as link prediction. We have highlighted this in our future work section (see Lines 534-535).
>
>
> **W3**:  We chose TER+AER as baselines because they are state-of-the-art methods for edge classification, capturing both structural features and higher-order proximities. Instead of comparing with other models, we focused on evaluating truss robustness against other edge-based features to better evaluate. We also included results for the geometric variant of TER+AER, along with AUC scores and ablation study results, in the appendix due to space constraints.
>
>
> We would also like to address your Assorted minor comments:
>
> **AMC1**: Thank you for your recommendation. Most of the notations are already listed in Table 3. However, we will ensure that any remaining ones are also included in the final version.
>
> **AMC2**: Thanks for noticing. This has been updated in the revised PDF.
>
> **AMC3**: We use $E$ to represent the edges of $G$ and $E(e, G)$ to denote the set of edges incident to $e$. These notations were not introduced by us but are adopted from previous studies to ensure consistency with existing work.
>
> **AMC4**: Thanks for pointing this out. This has been updated in the revised PDF.
>
>
>
> Now, we would like to address the question you asked,
>
> **Q1**: Please refer to our answer on W1 above.
>
> **Q2**: We selected TER+AER as baselines because they are the state-of-the-art methods for edge classification. Our truss robustness metrics can be integrated into any edge classification model, including those reported in Wang et al., 2023.
>
> **Q3**: We did not focus on hyperparameter tuning for PageRank and used the default damping factor of 0.85.

---

> > ### Author Response · Authors · 2024-11-30
> >
> > Thanks again for your valuable feedback. Could you please acknowledge that you read our response to your comments? If your concerns are addressed, we’d be grateful if you can adjust the score. If not, we’d love to engage further to address your comments.

---

> > > ### Comment · Reviewer_Hyhz · 2024-12-02
> > >
> > > Thanks to the author for the reply. I don't think those major concerns have been fully addressed and I will maintain the score.

---

> > > > ### Author Response · Authors · 2024-12-02
> > > >
> > > > Thank you for your response. We believe we have addressed all the comments and questions you have. **Please let us know if there are any specific points we may not have addressed thoroughly, and we will be happy to provide further clarifications.**

---

### Author Response · Authors · 2024-12-04

We sincerely thank all the reviewers for their valuable feedback and constructive suggestions. We have addressed all the concerns raised by the reviewers and updated our pdf accordingly (if necessary).


In summary, to address the reviewers' comments, we have provided new results that demonstrate the applicability of our approaches. The key additions are:
- We have provided the p-values and Cohen's d from the t-test comparing the performance of the last two columns in Table 2 (coreness+degree vs. our metrics) for each graph. These results demonstrate that the performance improvements in our metrics are statistically significant and practically impactful.

- The primary contribution of our paper is improving the performance of state-of-the-art edge classifications. However, to address reviewer kpuv's comment for runtime comparisons with other baselines, we conducted additional experiments and provided the results. These results demonstrate that our metrics achieve comparable runtime performance to simpler graph properties, while our ERC algorithm is 37 times faster than the computation of RS_{OD} and RS_{ID}.

We are hopeful that our clarifications and the newly provided results will be carefully considered and reflected in the final scores.

---

### Note · Authors · 2025-02-21

I have read and agree with the venue's withdrawal policy on behalf of myself and my co-authors.

---

### Meta-Review · Area_Chair_kLic · 2024-12-20

**Metareview:**

The authors investigate edge-level robustness in truss decomposition. Specifically, they construct a dependency graph that captures the impact of each edge's deletion on its neighboring edges. Based on this graph, they define three edge-level robustness measures. Algorithmically, they propose an efficient method for constructing the dependency graph. On the application side, they demonstrate the utility of these measures for edge classification tasks.

The reviewers found the concept of truss robustness interesting and appreciated the theoretical soundness of the proposed algorithm.

However, they raised the following concerns:
- W1: The connection between the proposed concepts and their application to edge classification needs to be more explicitly established.
- W2: The experiments could be more extensive with additional tasks, backbone models, and competitors.
- W3: Runtime comparisons are missing.

While some concerns, including W3, were addressed during the discussion period, the paper still has substantial room for improvement. The meta-reviewer recommends that the authors revise their work based on the provided feedback and submit it to a future conference.

**Additional Comments On Reviewer Discussion:**

Despite the discussion between the authors and reviewers, several concerns remained unresolved.

---

### Decision · Program_Chairs · 2025-01-22

Reject